# Immunostimulatory Activity of *Cordyceps militaris* Fermented with *Pediococcus pentosaceus* SC11 Isolated from a Salted Small Octopus in Cyclophosphamide-Induced Immunocompromised Mice and Its Inhibitory Activity against SARS-CoV 3CL Protease

**DOI:** 10.3390/microorganisms10122321

**Published:** 2022-11-23

**Authors:** Kyu-Ree Dhong, Ha-Kyoung Kwon, Hye-Jin Park

**Affiliations:** Department of Food Science and Biotechnology, College of BioNano Technology, Gachon University, 1342 Seongnam-daero, Sujeong-gu, Seongnam-si 13120, Republic of Korea

**Keywords:** *Cordyceps militaris* grown on germinated *Rhychosia nulubilis* fermented with *Pediococcus pentosaceus* SC11 (GRC-SC11), immune stimulatory, cyclophosphamide, SARS-CoV 3CL protease, splenocyte proliferation, macrophage, phagocytic activity

## Abstract

In this study, we investigated the immune-enhancing and anti-viral effects of germinated *Rhynchosia nulubilis* (GRC) fermented with *Pediococcus pentosaceus* SC11 (GRC-SC11) isolated from a salted small octopus. The cordycepin, β-glucan, and total flavonoid contents increased in GRC after SC11 fermentation. GRC-SC11 inhibits 3CL protease activity in severe acute respiratory syndrome-associated coronavirus (SARS-CoV). GRC-SC11 significantly increased thymus and spleen indices in immunocompromised mice. The rate of splenocyte proliferation was higher in GRC-SC11-treated immunocompromised mice than that in GRC-treated immunocompromised mice in the presence or absence of concanavalin A. In addition, GRC-SC11 increased the phagocytic activity and nitric oxide production in immunocompromised mice. The mRNA expression of interferon-gamma (IFN-γ), interferon-alpha (IFN-α), and interferon-stimulated gene 15 (ISG15) was up-regulated in GRC-SC11 treated RAW 264.7 macrophages, compared to GRC. Our study indicates that GRC-SC11 might be a potential therapeutic agent for immunocompromised patients who are vulnerable to SARS-CoV infection.

## 1. Introduction

The immune system protects the body from infectious agents, such as bacteria and viruses, by sensing and removing them. Immunocompromised patients are those with a compromised immune system owing to immunodeficiency (e.g., aging, cancer, etc.,) or immunosuppressive therapy. These immunocompromised patients do not respond normally to infections due to a destroyed or weakened immune system and are more vulnerable to viral infection than the healthy population. Baek et al. reported that immunocompromised patients with COVID-19 have a significantly higher in-hospital mortality rate than non-immunocompromised patients with COVID-19 (9.6% vs. 2.3%) [1]. Many countries have tried to mitigate the risk of COVID-19 among immunocompromised patients by offering vaccines and adjuvant agents. CY is an anti-cancer drug. However, undesirable side effects were reported because the CY uptake into normal cells happens more compared to in cancer cells [2]. Excessive CY treatment (about 100 mg/kg) in mice could lead to immunosuppression by decreasing the proliferation and phagocytic activity of macrophages and killing normal cells [3,4,5]. CY also can induce a reduction in the number of T cells and B cells [6]. Therefore, many research groups treat high doses of CY to mice to create an animal model with immunocompromised condition [7,8,9].

The human immune system consists of innate and adaptive systems. In the innate immune system, immune cells (macrophages, dendritic cells, neutrophils, and natural killer cells) directly recognize infectious agents via receptors. Antigen-presenting cells, such as macrophages, dendritic cells, and B cells, recognize, process, and present antigens to the lymphocytes. In addition, macrophages secrete various pro-inflammatory cytokines. The adaptive immune system includes T lymphocyte-mediated cellular and B lymphocyte-mediated humoral immunity. Humoral immunity involves antibodies that bind to antigens and block the infection of the host cell. In contrast, T-cell-mediated cellular immunity is vital for the prevention of infections such as viral and bacterial via recognizing and destroying pathogen-infected cells. Imbalance of the immune system in immunocompromised patients can aggravate infectious, inflammatory, and autoimmune diseases. Several studies have reported the medicinal uses of medicinal plants such as mushrooms owing to their anti-viral, immunomodulatory, anti-allergic, and anti-inflammatory effects [10,11]. Mushrooms, including *Phellinus linteus*, *Cordyceps militaris*, *Flammulina velutipes,* and *Grifola frondosa* have various effects, including anti-viral, anti-tumor, immunomodulating, and anti-inflammatory [12,13,14,15,16,17]. Among the mushrooms, *Cordyceps militaris* is widely used because of its immunomodulatory and anti-viral effects [12,18].

*C. militaris*, an entomopathogenic fungus, has been used in traditional medicine to regulate human health in Korea, China, and Japan [19]. *C. militaris* is used to prevent infections and complications in immunocompromised patients [20]. Several studies have shown that *C. militaris* has anti-tumor, anti-inflammatory, and immunomodulatory activities [21,22,23]. It has also been reported that *C. militaris* induces T-cell proliferation, natural killer (NK)-cell-based cytotoxic activity, and Th1 cytokine secretion [24,25,26]. However, because of the low abundance of *C. militaris* in nature, we cultivated *C. militaris* on germinated *Rhynchosia nulubilis* (GRC). *C. militaris* grown on germinated soybeans (GSC) exhibits immune stimulation [27], anti-cancer [28], and anti-allergic activities [29,30]. Additionally, polysaccharides from GSC showed anti-influenza and anti-viral activity that activates macrophages and increases the expression of cytokines (e.g., TNF-α, IFN-γ, and IL-6) [31]. However, *C. militaris* or *C. militaris* grown on GSC requires an expensive processing procedure for extraction and to exert a good biological effect. In this study, we fermented GRC with lactic acid bacteria (LAB) to overcome these issues. LAB belonging to the order *Lactobacillales* that produce lactic acid include *Lactobacillus*, *Enterococcus*, *Pediococcus*, and *Streptococcus*, which are Gram-positive anaerobes [32,33]. Previous studies have demonstrated that these LABs could be used to enhance immune system activity by providing protection against invading pathogens and maintaining good intestinal health by regulating the intestinal microbial population [34]. To the best of our knowledge, no studies have reported the anti-viral and immunoregulatory effects of GRC fermented with *Pediococcus pentosaceus* SC11 (GRC-SC11). This study aimed to investigate whether GRC-SC11 could exert both antiviral and immunostimulatory effects. This is the first study to investigate the anti-viral and immunostimulatory effects of *C. militaris* fermented with LAB using a 3CL protease activity assay and to check the activity and population of immune cells in cyclophosphamide (CY)-treated immunocompromised mice.

## 2. Materials and Methods

### 2.1. Preparation of GRC Fermented with Probiotic Strains

GRC extracts were prepared using patented technologies developed by the Cell Activation Research Institute (CARI, Seoul, Korea) where a voucher specimen was deposited (Kucari:0903). *P. pentosaceus* isolated from salted small octopus (SC11) and onion (ON188), *Lactobacillus pentosus* from pickled burdock (SC64 (voucher specimen number Gaucari:0802)), *Lactococcus lactis* from broccoli (Bro17 (voucher specimen number Gaucari:0806)), and *Weissella cibaria* from sweet pumpkin (S. pum8 (voucher specimen number Gaucari:0807)) strains used in this study were obtained from Dr. YS Park. For fermentation, GRC extracts were prepared by adding distilled water to GRC in the ratio 20:1 (e.g., 20 mL of distilled water per 1 g GRC) followed by heating at 105 °C for 2 h. GRC extracts were centrifuged at 13,100× *g* for 10 min, and the supernatant was fermented with LAB. Isolated LABs (*P. pentosaceus* SC11, *P. pentosaceus* ON-188, *L. pentosus* SC64, *L. lactis* Bro17, and *W. cibaria* S. pum8) were inoculated into the extracts and incubated at 37 °C for 24 h. After fermentation, the fermented products were separated by centrifugation at 13,100× *g* for 10 min. GRC-inoculated probiotics were heat-killed at 100 °C for 10 min and sonicated for 3 min (Sonics & Materials Inc., Washington, DC, USA) [35].

### 2.2. Measurement of Cordycepin in GRC-SC11

The amount of cordycepin in GRC-SC11 was determined by UPLC quantitative analysis, as described previously [8]. UPLC-Q-TOF-MS analysis was performed using a Waters Acquity UPLCTM system (Waters Corp., Milford, MA, USA) equipped with a Waters Acquity BEH C18 column (100 mm × 2.1 mm × 1.7 μm particle size; Waters Corp.) and Waters Q-TOF Premier (Micromass MS Technologies, Manchester, UK). The UPLC-Q-TOF-MS analysis was performed following previously described methods [8].

### 2.3. Measurement of β-Glucan in GRC-SC11

As previously described [8], the amount of β-glucan in mushroom preparations was analyzed using the Megazyme K-YBGL kit (Megazyme, Bray, Ireland). The experiment was conducted according to the protocol provided by Megazyme International, Ireland.
β-glucan (% *w*/*w*) = total glucan (% *w*/*w*) − α-glucan (% *w*/*w*)

To measure the total glucan content, the sample was mixed with 12 M H_2_SO_4_ on ice for 2 h and water was added. After vigorous stirring, the tube was opened slightly and placed in boiling water at 100 °C for 2 h. The sample and 6 mL of 10 M KOH solution were added to a 100-mL volumetric flask and adjusted to volume with 200 mM sodium acetate buffer (pH 5). The mixed sample was aliquoted into a polypropylene centrifuge tube and centrifuged at 1500× *g* for 10 min. Exo-1,3-β-glucanase and β-glucosidase were mixed with the tube contents on a vortex mixer and incubated at 40 °C for 60 min. The GOPOD reagent was added to each tube and incubated at 40 °C for 20 min. The absorbance of all the samples was measured at 510 nm using a microplate reader (Epoch, Biotek Instruments, Inc., Winooski, VT, USA).

To measure α-glucan, the sample was mixed with 2 M KOH and 1.2 M sodium acetate buffer (pH 3.8) and added to 0.2 mL amyloglucosidase (1630 U/mL) and invertase (500 U/mL). After incubating and mixing at 40 °C for 30 min, 0.1 mL sodium acetate buffer (200 mM, pH 5.0) and 3 mL GOPOD reagent were added to the aliquoted samples. The mixture was incubated at 40 °C for 20 min, and the absorbance of all samples was measured at 510 nm using a microplate reader (Epoch).

### 2.4. Total Flavonoid Content Assay

The total flavonoid contents of GRC-SC11 and GRC were measured as previously described [36]. Fifty microliters each of GRC-SC11 and GRC and 15 μL of 5% sodium nitrite solution were mixed and incubated for 5 min. Fifteen microliters of 10% aluminum nitrate solution were added. After 10 min at room temperature in the dark, 100 μL of 1N sodium hydroxide solution was added to the solution for 11 min. Absorbance was measured at 510 nm using a microplate reader (Epoch, Biotek Instruments, Inc., Winooski, VT, USA). Quercetin was used as the standard, and the results are represented as milligrams of QE per gram (mg QE/g) of GRC-SC11 and GRC.

### 2.5. CL Protease Activity (SARS-CoV-2 3CLPro)

For screening the activity of 3CL protease, 3CL Protease Assay Kit (#78042-1, BPS Bioscience, San Diego, CA, USA) was used. Stock solutions (100 µg/mL) of GRC-SC11 and GRC extracts were prepared. About 5 ng of 3CL protease enzyme was added to each well except “Blanks”. GC376 (500 μm), a 3CL protease inhibitor control, was distributed in the allocated wells, and 1X assay buffer solution was added. After adding 250 μm of 3CL protease substrate to each well, the mixtures were incubated at room temperature for 4 h. Fluorescence was measured using a microplate reader at 360 nm excitation and 460 nm emission. The inhibition activity was calculated according to the protocol of Durdagi et al. [37].

### 2.6. Cell Culture of RAW264.7 Macrophages

Murine RAW 264.7 cells (ATCC^®^ TIB-71™) were obtained from the American Type Culture Collection (ATCC, Manassas, VA, USA) and grown in Dulbecco’s modified Eagle’s medium (DMEM) (Welgene, Seoul, Korea) supplemented with 1% penicillin (10,000 U/mL)/streptomycin (10,000 U/mL) (Welgene) and 10% heat-inactivated fetal bovine serum (FBS) (Welgene). The cells were grown at 37 °C in a humidified 5% CO_2_ atmosphere.

### 2.7. Quantification of Nitric Oxide (NO) Production

The total amount of NO produced was investigated by the Griess reaction assay, as described previously [8]. RAW 264.7 cells were seeded in 96-well plates (2 × 10^5^ cells/well) overnight and then stimulated with samples ((GRC-*P. pentosaceus* SC11 (GRC-SC11), GRC-*L. pentosus* SC64 (GRC-SC64, voucher specimen number Gaucari:0802), GRC-*L. lactis* Bro17 (GRC-Bro17, voucher specimen number Gaucari:0806), and GRC-*P. pentosaceus* ON188 (GRC-ON188; voucher specimen number Gaucari:0805), GRC-*W. cibaria* S. pum8 (GRC-S. pum8, voucher specimen number Gaucari:0807)) for 48 h. Primary cultured peritoneal macrophages from CY-treated mice or normal mice were seeded in 96-well plates (2 × 10^5^ cells/well) and incubated with GRC-SC11 and GRC for 48 h. Next, 100 μL of the culture supernatant was incubated and mixed at room temperature for 10 min with Griess reagent (1% sulfanilamide, 0.5% phosphoric acid, and 0.1% naphthylenediamine dihydrochloride). Absorbance was measured at 540 nm using a microplate reader (Epoch).

### 2.8. Measurement of the Cell Viability Using Cell Counting Kit-8 (CCK-8)

Cell viability was measured using CCK-8 (DOJINDO Laboratories, Kumamoto, Japan), as described previously [8,38,39]. RAW 264.7 cells were plated onto 96-well plates (2 × 10^5^ cells/well) and incubated in the presence or absence of GRC-SC11, GRC-SC64, GRC-Bro17, GRC-ON188, GRC-spum8, and GRC for 48 h (Figure 1). After adding the CCK-8 solution, cell viability was measured using a microplate reader at 450 nm (Epoch). Primary peritoneal macrophages isolated from CY-treated immunosuppressed mice were plated in 96-well plates (5 × 10^4^ cells/well). The cells were incubated with GRC and GRC-SC11 for 24 h, CCK-8 solution was added, and the absorbance was measured at 450 nm.

### 2.9. Animals and Experimental Design

Female 6–8-week-old BALB/c mice with a body weight of 18 ± 2.0 g were purchased from Daehanbiolink Co., Ltd. (Seoul, Korea) and acclimatized to our laboratory conditions for 7 days before commencement of the animal experiment. The mice were housed in an air-conditioned room at 25 °C and 40–60% humidity. All mice were provided free access to standard sterile water and diet and were randomly divided into four groups as follows: Control, Control-CY, GRC-SC11-CY, and GRC-CY. Each group was fed daily with the indicated heat-inactivated sample at doses of 1.0 × 10^8^ colony forming units (CFU)/mL/mouse. The control group was fed with 0.1 mL of distilled water. On day 20, mice from each group were sacrificed for experiments. All animal experiments were performed in accordance with the instructions of the Medical Ethics Committee for the Use of Experimental Animals at Gachon University (GIACUC-R2015014).

### 2.10. Isolation and Cell Cultures of Splenocytes from Murine Model

Spleens were obtained from CY-treated mice sacrificed under aseptic conditions, washed with RPMI 1640 medium, and crushed to isolate splenocytes. Splenocytes were passed through a 200-mesh stainless-steel sieve to obtain a homogeneous cell suspension. The splenocyte suspension was washed twice with RPMI 1640-FBS medium and centrifuged at 200× *g* for 7 min. The recovered splenocytes were resuspended in red blood cell lysis buffer (155 mM NH_4_Cl, 12 mM KHCO_3_, and 0.1 mM ethylenediaminetetraacetic acid (EDTA) solution, pH 7.2) for 4 min to remove red blood cells. After centrifugation, the harvested splenocytes were resuspended in the RPMI 1640 medium. Splenocytes were grown in RPMI 1640 at 37 °C under a fully humidified 5% CO_2_ atmosphere.

### 2.11. The Mitogenic and Co-Mitogenic Activity Assay in Primary Splenocytes from CY-Treated Immunosuppressed Mice

Splenocytes were seeded into 96-well plates (2 × 10^5^ cells/well). In the mitogenic and co-mitogenic activity assays, GRC or GRC-SC11 was added in the presence or absence of concanavalin A (Con A) (5 μg/mL). After 24 h of incubation, the CCK-8 solution was added to each well and incubated for 4 h. Absorbance was measured at 450 nm using a microplate reader (Epoch). The direct mitogenic effect of the tested compounds was expressed as a stimulation index of mitogenic = mean A450 test compound/mean A450 control, and the co-mitogenic effect was expressed as the stimulation index of co-mitogenic = mean A450 (Con A + test sample)/mean A450 Con A. Each substance was tested minimally in at least three independent experiments.

### 2.12. Splenic and Thymic Indices

The body weights of the mice were recorded. The spleen and thymus were excised from the mice and weighed immediately. Splenic and thymic indices were expressed using the following equation, as described previously [8,40].
Splenic and thymic indices = [spleen or thymus weight (mg)/body weight (g)]

### 2.13. Isolation and Cell Cultures of Primary Peritoneal Macrophage from Murine Model

Primary peritoneal macrophages were obtained from mice according to previously described methods [8]. Mice were sacrificed and peritoneal fluid was obtained by intraperitoneal injection of PBS. The peritoneal fluid solution was centrifuged at 377× *g* for 10 min. The deposited cell pellets were suspended in Roswell Park Memorial Institute (RPMI) 1640 (Welgene) medium supplemented with 10% FBS and 1% penicillin (10,000 U/mL)/streptomycin (10,000 U/mL). Isolated splenocytes were incubated for 4 h at 37 °C with 5% CO_2_ and washed with PBS to remove non-adherent cells.

### 2.14. Determination of Phagocytic Activity

Peritoneal macrophages were plated in 96-well plates (2 × 10^4^ cells/well) and incubated for 24 h in 5% CO_2_ at 37 °C. Each group was then treated with latex bead-Rabbit IgG-FITC complex (Cayman, MI, USA). To evaluate whether GRC-SC11 affected the phagocytic activity of fluorescent particles in primary cultured peritoneal macrophages, we used a real-time microscope (Nikon ECLIPSE Ti, Tokyo, Japan). Random regions were photographed, and live imaging software snapped a photo for 48 h. Imaging software files were exported and analyzed using MetaMorph software version 7.8.9.0 (Molecular device, Sunnyvale, CA, USA).

### 2.15. Reverse Transcriptase-Polymerase Chain Reaction (RT-PCR)

Total RNA was isolated from RAW 264.7 macrophages as previously described (Oh et al., 2011; D. K. Park, Choi, et al., 2012). The RT-PCR conditions were as follows: initial heat activation at 95 °C for 900 s, a denaturation step at 94 °C for 30 s, an annealing step at 57 °C for 90 s (IFN-γ), 59.9 °C for 90 s (TNF-α), or 58.1 °C for 90 s (GAPDH); an extension step at 72 °C for 60 s for 35 cycles; and a final extension at 60 °C for 1800 s. The following primers were used: *Mus musculus* IFN-γ forward 5′-GG TCT CAA CCC CCA GCT AGT-3′; *Mus musculus* IFN-γ reverse 5′-CA TGA TGC TCT TTA GGC TTT CCA G-3′; *Mus musculus* IFN-α forward 5′-TGG CTA GGC TCT GTG CTT TC-3′; *Mus musculus* IFN-α reverse 5′-CCT TCT TGA TCT GCT GGG CA-3′; *Mus musculus* ISG15 forward 5′-ATG GCC TGG GAC CTA AAG GT-3′; *Mus musculus* ISG15 reverse 5′-CTG GTC TTC GTG GAC TTG TTC-3′; *Mus musculus* GAPDH forward 5′-GCA AAG TGG AGA TTG CCA TC-3′; and *Mus musculus* GAPDH reverse 5′-CAT ATT TCT CGT GGT TCA CAC CC-3′ (Cosmo Genetech, Seoul, Korea). The bands were measured using the LI-COR Odyssey (LI-COR Biosciences Ins., Lincoln, NE, USA).

### 2.16. Statistical Analysis

The data are expressed as means ± standard deviation (SD) and were analyzed by one-way analysis of variance (ANOVA), Dunnett’s *t*-test, or Duncan’s *t*-test. All analyses were performed using SPSS software, version 12 (SPSS Inc., Chicago, IL, USA).

## 3. Results

### 3.1. Screening the Best Probiotics for Fermenting GRC with Enhanced Immune Stimulatory Activity

The released NO from macrophages affects the innate immune system, contributing to the killing of microorganisms or mediating biological functions as intracellular messenger molecules [27,41,42]. To investigate the effect of GRC fermented with probiotics on NO production in RAW 264.7 macrophages, we examined the amount of NO released after treating GRC fermented with selected probiotic candidate strains in RAW 264.7 macrophages. As shown in Figure 1, the immune regulatory effects of 10 isolated GRC-fermenting LAB strains (SC11, ON188, Bro17, spum8, and SC64) were compared to those of GRC. GRC fermented with SC11 (GRC-SC11) significantly increased NO production compared to the control (*p* < 0.001, Figure 1A) and increased RAW 264.7 proliferation compared to the control (*p* < 0.001, Figure 1B). Therefore, GRC-SC11 was chosen as the test sample for further study.

### 3.2. The Changed Content of Bioactive Compounds in GRC before and after Fermentation with P. pentosaceus (SC11)

To measure the changes in the content of physiologically active substances in GRC after fermentation with LAB, cordycepin and β-glucan contents were analyzed. Cordycepin of GRC-SC11 (10.45 ± 2.73 mg/g) was 1.4-fold higher than that of GRC (6.35 ± 1.64 mg/g) and β-glucan of GRC-SC11 (28.78 ± 0.31 g/100 g) was determined by mushroom β-glucan assay kit to be 2.5-fold higher compared to that of GRC (11.81 g/100 g) (Table 1). Total flavonoid content of GRC-SC11 (6.83 ± 1.98 mg QE/g extract) was 2.4-fold higher than that of GRC (2.8 ± 0.85 mg QE/g extract).

### 3.3. Inhibitory Activity of GRC-SC11 against 3CL Protease

Many studies have demonstrated that cordycepin and β-glucan possess anti-viral activity [43]. To investigate the anti-viral effect of GRC-SC11, we examined whether it could inhibit the activity of the severe acute respiratory syndrome-associated coronavirus 2 (SARS-CoV-2) 3CL protease, which is essential for virus replication through the processing of viral polyproteins. Inhibitory concentration (IC_50_) value of GRC-SC11 on inhibited 3CL protease activity is 224.33 µg/mL, while that of GRC is 533.78 ug/mL (Figure 2, *p* < 0.05). GRC-SC11 significantly decreased the 3CL protease activity, compared to GRC and control (*p* < 0.001 vs. con.). These results suggested that GRC-SC11 inhibits the activity of SARS-CoV-2 main protease (3CL protease).

### 3.4. Effect of GRC-SC11 on Splenic and Thymic Indices in Immunocompromised Mice

As shown in Figure 3, the spleen and thymus indices of the CY-treated groups were significantly reduced compared with those of the untreated groups (*p* < 0.05), suggesting that the immunocompromised model was successfully created. The spleen and thymus indices of the GRC-SC11-treated group in the CY-treated immunocompromised murine model were significantly higher than those in the CY-treated groups (*p* < 0.001, Figure 3).

### 3.5. The Mitogenic and Co-Mitogenic Effects of GRC-SC11 on Primary Splenocytes from CY-Induced Immunocompromised Murine Model

We evaluated the proliferation of splenocytes from CY-treated mice to determine the mitogenic effect of GRC-SC11 on the immune system. As shown in Figure 4, GRC-SC11 significantly increased the proliferation of splenocytes compared to the GRC and control groups (*p* < 0.01). In addition, GRC-SC11 showed significant co-mitogenic activity in Con A-stimulated lymphocytes compared to GRC and Con A (Figure 4). Thus, GRC-SC11 might exert a mitogenic effect on mouse splenocytes and a co-mitogenic effect on Con A-treated murine splenocytes.

### 3.6. Effects of GRC-SC11 on Phagocytic Activity of Peritoneal Macrophages from CY-Treated Immunocompromised Mice and on the Levels of INF-γ mRNA Expression

To evaluate whether GRC-SC11 affected the phagocytic activity of primary cultured peritoneal macrophages, phagocytic activity was assessed by measuring the fluorescence intensity of internalized IgG-opsonized fluorescein isothiocyanate (FITC) particles in primary peritoneal macrophages isolated from CY-treated immunosuppressed mice. The phagocytosis-promoting activity of GRC-SC11 was significantly higher than that of GRC and control groups (*p* < 0.05, Figure 5A). As shown in Figure 5B, the phagocytic index of peritoneal macrophages from the GRC-SC11-treated groups was significantly higher than that from the GRC control group and CY-treated control group (*p* < 0.05). After GRC-SC11 treatment, the immune activity of macrophages, in which the immune response was significantly reduced, affected phagocytic activity, acting as a nonspecific immune defense.

The expression of interferon-gamma (IFN-γ), interferon-alpha (IFN-α), and interferon stimulated gene 15 (ISG 15) mRNA were significantly increased in the GRC-SC11-treated group compared to that in the GRC and control groups (*p* < 0.001, *p* < 0.01 and *p* < 0.05) (Figure 5D).

## 4. Discussion

Immunocompromised patients are more susceptible to viral infections and progress to severe malignant conditions [44]. In recent years, more than 30 million people have suffered from the COVID-19 infection, and more than 1 million people have died worldwide. Among COVID-19 casualties are immunocompromised patients with immunodeficiency syndrome (AIDS), cancers, organ transplantation, and immunosuppressive medications [45]. There are several vaccines and drugs available against this outbreak; however, formidable risks and side effects need to be resolved. Recently, many researchers have focused on natural products, including mushrooms and their derived compounds, because they possess direct anti-COVID-19 activities or adjuvant effects that boost the immune system. In this study, we investigated whether GRC after fermentation with LAB could enhance its biological activities. Probiotics isolated from traditional fermented foods have various health-promoting effects, including immunomodulation, regulation of intestinal microflora balance, antagonistic effects against pathogens, and anti-viral effect [46,47]. It has been reported that fermentation by LAB can enhance flavor, safety, and physiological activity, increase the digestion efficiency of natural substances, and produce biologically active metabolites [48]. Thus, the immunostimulatory activity of GRC fermented with *P. pentosaceus* SC11 (GRC-SC11) in immunocompetent models were evaluated in this study.

To investigate whether GRC-SC11 changed the profile of physiological substances that possess immune-enhancing activity, analysis of bioactive compounds such as cordycepin, β-glucan, and TFC in GRC-SC11 and GRC was performed. The content of cordycepin of GRC-SC11 was significantly increased, compared to that of GRC. Cordycepin (3′-deoxyadenosine) has been reported to have immune-enhancing activity in cyclophosphamide-induced immunosuppressed mice [49,50]. The relationship between the mechanisms of cordycepin synthesis and LAB fermentation has not yet been elucidated. However, the production of cordycepin through lactic acid fermentation may be similar to the synthesis of 2-deoxrtivonucleotides from adenosine, adenosine monophosphate, or adenosine diphosphate by ribonucleotide reductase [51,52,53]. Recent studies have reported that cordycepin showed anti-viral activity by targeting the spike protein, main protease, and RNA-dependent RNA polymerase of viruses [54,55]. The β-glucan content of GRC significantly increased after SC11 fermentation (*p* < 0.01, Table 1). The common *C. militaris* contains β-(1,3)-glucan, which exerts immunostimulatory effects on natural killer cells, splenic lymphocytes, and THP1 monocytes [20,27,56,57,58,59,60]. Additionally, β-glucan exhibit anti-viral activity by increasing the expression of IFN-γ and NO in bronchoalveolar lavage fluid and macrophages [61,62]. IFN-γ can inhibit viral replication by suppressing viral genome synthesis and expression of late gene proteins [63]. NO inhibits microbial replication by regulating DNA synthesis [64]. Some studies have reported that *P. pentosaceus* contains polysaccharide-degrading enzymes such as β-glucosidase, which can produce deglycosylated compounds from glycosylated precursors, thereby enhancing the nutritional value and bioavailability of plant metabolites [65,66,67]. It is suggested that the polysaccharide chain may be cleaved by degrading enzymes during LAB fermentation to produce low molecular weight polysaccharides, which are more easily absorbed from the small intestine [67]. Therefore, these data suggest that fermentation of GRC with LAB could improve biological activities such as immunostimulatory and anti-viral activities by increasing bioactive compounds and changing their chemical structures and physiological characteristics (Appendix A).

The innate immune system, the first line of defense, senses the virus through pattern recognition receptors and stimulates signaling pathways that contribute to viral clearance [68]. Among the cells in the innate immune system, macrophages are antigen-presenting cells that initiate defense processes against external pathogens such as foreign cells, bacteria, or microorganisms through phagocytosis [27,69,70]. The phagocytosed particles are digested through endosomes and lysosomes containing a range of proteases that degrade the particles [71,72]. NO can be used as a quantitative indicator of macrophage activation and is a physiological molecule for killing pathogens and regulating cell functions, as it acts as a short-lived messenger [42,73]. Activated macrophages produce NO to protect cells against the intrusion of foreign substances [41]. To explain the immunostimulatory activity of GRC fermented with various LAB strains on macrophages, we examined NO production by RAW264.7 macrophages. GRC-SC11 significantly increased the amount of released NO compared to the untreated control (Figure 1). This result suggests that GRC-SC11 can enhance immune activity by increasing macrophage activity.

Additionally, we investigated the anti-viral activity of GRC-SC11 that might rescue the immunocompromised patients infected with COVID-19. Verma reported that cordycepin from *Cordyceps militaris* showed anti-viral activity against SARV-CoV-2 by binding 3CL protease [55]. Khan et al. reported that flavonoid showed anti coronavirus activity by binding to the main protease (3CL protease) and decreasing the cytopathic effect induced SARS-CoV-2 [74,75]. Seri et. al. reported that flavonoid showed the antiviral activity by binding to S site of 3CL protease due to hydrophobic aromatic rings and hydrophilic hydroxyl groups of flavonoids [76]. Arunkumar et al. demonstrated that beta glucan could be treatment for SARV-CoV-2 due to the strong affinity with 3CL protease in silico [77]. Ohta el al. demonstrated the anti-viral activity of the acidic polysaccharide, Araf-(1→, →5)-Araf-(1→, →4)-Galp-(1→ and →4)-GalAp-(1→) residues isolated from hot-water extracts of *Cordyceps militaris* grown on germinated soybeans (GSC) by enhancing immuno-stimulating activity [31]. In addition, previously identified novel flavonoids (daidzein 7-O-β-d-glucoside 4″-O-methylate (CDGM), glycitein 7-O-β-D-glucoside 4″-O-methylate (CGLM), genistein 7-O-β-D-glucoside 4″-O-methylate (CGNMI) and genistein 4′-O-β-D-glucoside 4″-O-methylate (CGNMII)) in GSC, showed biological activities, including anti-allergic and immunostimulating activity [78] [22]. Isoflavonoids showed potential antiviral activity by binding with ACE-2 and M pro receptors [79]. Interestingly, we observed that the contents of cordycepin, total flavonoid and β glucan were all increased in GRC fermented with *Pediococcus pentosaceus* SC11, compared to GRC. Recently, it is reported that vitamin D in the *Cordyceps militaris* exerted the antiviral activity by inhibiting SARS-CoV-2 attachment to host cells and increasing the expression of interferon beta [80,81]. Probiotics eradicate the enteric viruses [47]. This implies that the presence of vitamin D and probiotics in GRC-SC11 may also contribute to its anti-viral activity. To alleviate the COVID-19 symptoms, it is crucial to enhance host immune system as well as suppress 3CL protease activity in SARS-CoV-2, which is essential for virus replication through processing the viral polyproteins from SARS-CoV-2 RNA, which is critical for virus replication [82,83]. Therefore, to determine the anti-viral activity of GRC-SC11, we examined whether GRC-SC11 could affect the activity of the SARS-CoV-2 3CL protease. We showed that GRC-SC11 significantly inhibited 3CL protease activity compared to GRC (Figure 2). In accordance with our previous data, the increased content of bioactive compounds in GRC after SC11 fermentation might be responsible for the stronger anti-viral activity through inhibiting 3CL protease compared to GRC. It is worth tracking the active compounds for targeting 3CL protease in viruses in the future study.

Many studies have indicated that viral infections are more fatal in immunocompromised patients [1]. Thus, we examined whether GRC-SC11 recovered immune activity in an immunocompromised animal model. Cyclophosphamide (CY) is one of the most frequently used anticancer and immunodepression drugs [7,84,85]. Patients administered chemotherapeutic agents such as CY often exhibit immunosuppressive activities such as decreased peripheral blood leukocytes, granulocytopenia, or neutropenia. These patients with rapidly diminished immunity are easily exposed to infections or diseases because of the inactivation of immune cells [86,87]. In the current study, high-dose CY was orally administered to mice to mimic immunocompromised patients, and we found that orally administered GRC-SC11 recovered immune organ weights, including the spleen and thymus, in CY-treated immunosuppressed mice (Figure 3). The immune organ index reflects immune capabilities [88]. Therefore, GRC-SC11 might induce the differentiation and proliferation of T and B cells in the thymus and spleen, which could protect from viral infections. To observe the mitogenic effect of GRC-SC11 on T cell, GRC-SC11 was treated with Con A, a plant lectin isolated from *Canavalia ensiformis* (jack bean) seeds to induce the mitogenic activity of T lymphocytes present in primary splenocytes from mice [89]. The present study indicated that GRC-SC11 alone or in combination with Con A significantly increased the T-cell population in primary splenocytes, suggesting that GRC-SC11 enhanced the T-cell-mediated adaptive immune system in CY-treated immunocompromised mice (Figure 4). Subsequently, we examined whether GRC-SC11 could stimulate innate immunity in an immunocompromised murine model.

Macrophages play a central role in innate immunity to initiate defense processes against external pathogens such as foreign cells, bacteria, or microorganisms through phagocytosis [27,69,70]. The phagocytosed particles are digested through endosomes and lysosomes containing a range of proteases that degrade the particles [71,72]. The IgG-opsonized FITC particles are taken up by FcγR or TLR receptors on macrophages, and they develop into phagosomes and fuse with lysosomes [8,90]. In addition, the amount of NO released and phagocytic activity of primary peritoneal macrophages from CY-treated mice were assessed. We found that GRC-SC11 significantly increased the levels of released NO compared to the GRC and CY-treated controls without cytotoxicity (Figure 2A,B). Consistent with previous reports, CY significantly impaired phagocytosis and the number of primary peritoneal macrophages isolated from CY-treated mice compared to that from untreated control group [8,91,92,93,94]. We observed that the phagocytic index of GRC-SC11-treated group was significantly increased in the immunosuppressed primary peritoneal macrophages as compared to that of the GRC-treated group (Figure 5). These results suggested that GRC-SC11 may contribute to immunity enhancement through an increase in NO production and activation of phagocytic activity in macrophages.

We investigated whether GRC-SC11 could affect cytokine production in macrophages, which may enhance immune activity or anti-viral activity. Our results showed that GRC-SC11 increased the level of IFN-γ, IFN-α and ISG15 in RAW 264.7 macrophages. IFN-γ is a key effector in cellular immunity that inhibits viral infection and enhances antigen processing and presentation [95]. IFN-γ can also block viral replication through RNA synthesis. IFN-γ induces Janus kinase (JAK)/signal transducers and activators of transcription (STAT) signaling pathway and activates the transcription of IFN-γ stimulated genes (ISG) such as IFN regulatory factor (IFN). ISG inhibits viral replication by inducing viral RNA degradation and inhibiting viral translation [96]. IFN-γ signaling upregulates the expression of MHC I complexes, thereby increasing the recognition of pathogen-derived antigens by effector T cells. IFN-γ also activates NO synthases by inducing inducible isoform nitric oxide synthase (iNOS) and increases the expression of proinflammatory molecules such as TNF-α [97,98]. Therefore, the results suggest that GRC-SC11 might induce anti-viral activity and enhance the immune response. However, whether GRC-SC11 has the same effect on cancer-induced murine models as CY treatment and whether the glycosylated isoflavonoid compounds of GRC-SC11 is converted to the aglycone needs to be further investigated in future studies.

## 5. Conclusions

In conclusion, GRC-SC11 exerts both anti-viral and immunostimulatory effects in vivo and in vitro. The contents of β-glucan, cordycepin, and total flavonoids were higher in GRC after *P. pentosaceus* SC11fermentation than that in GRC. GRC-SC11 inhibited 3CL protease activity of SARS-CoV, but increased IFN-γ production in macrophages and proliferation of T lymphocytes and splenocytes from CY-treated immunosuppressed mice. The enhanced immunostimulatory and anti-viral activities of GRC-SC11 might be due to the increased content of bioactive compounds in GRC after SC11 fermentation. GRC-SC11 also enhances macrophage activity by increasing NO production and phagocytic activity; the macrophages which were isolated from immunocompromised mice. Increased immune response can protect immunocompromised patients from infectious viral diseases. Taken together, our results suggest that GRC-SC11 may be used as a therapeutic agent for patients or children with weakened immune systems who are more susceptible to SARS-CoV infection.

## Figures and Tables

**Figure 1 microorganisms-10-02321-f001:**
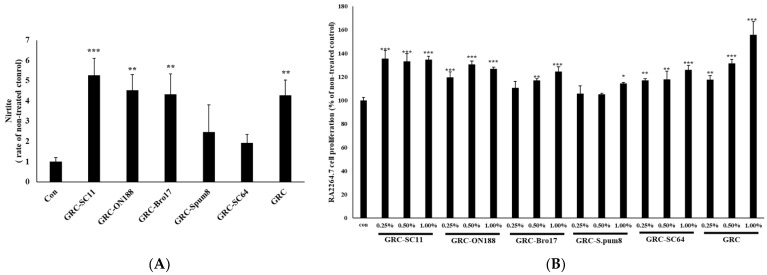
Effect of GRC fermented with different lactic acid bacteria stains on nitric oxide (NO) production (**A**) RAW264.7 cells were treated with GRC-SC11, GRC-SC64, GRC-Bro17, GRC-ON188, GRC-S. pum8, GRC. Nitrite levels in the culture media were determined using Griess reagent and were presumed to reflect NO levels. (**B**) RAW264.7 cells were treated with various concentrations (0.25%, 0.50%, and 1.00%) of GRC-SC11, GRC-ON188, GRC-Bro17, GRC-S. pum8, GRC-SC64, GRC for 24 h. RAW264.7 cell viability was assessed using CCK-8 assay. One-way analysis of variance was used for the comparison of group means, followed by Dunnett’s *t*-test (* *p* < 0.05, ** *p* < 0.01, *** *p* < 0.001 vs. control (con)).

**Figure 2 microorganisms-10-02321-f002:**
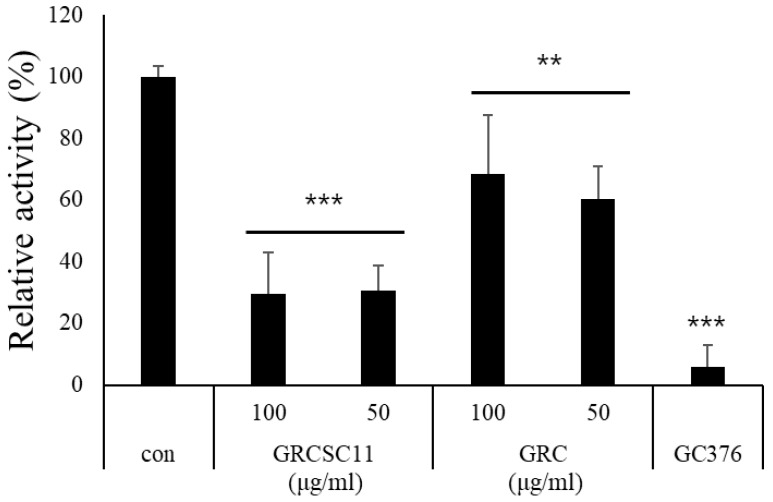
Inhibition of 3CL protease activity of GRC-SC11 and GRC. The results are shown as the mean ± SD (*n* ≥ 8 animals in each group). GC376 is a 3CL inhibitor used as a positive control. Significance of the results was determined using one-way analysis of variance followed by Dunnett’s *t*-test (*** *p* < 0.001 and ** *p* < 0.01 vs. con).

**Figure 3 microorganisms-10-02321-f003:**
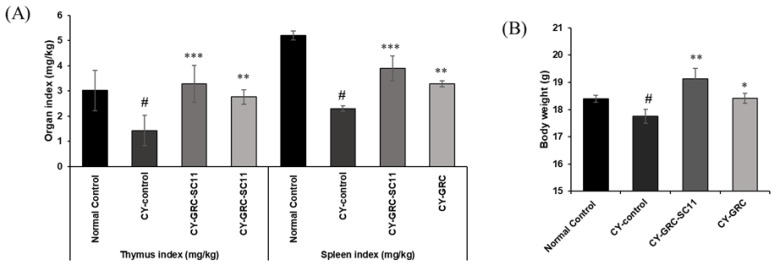
Effects of GRC-SC11 on immune organs in CY-treated immunocompromised mice. Both GRC and GRC-SC11 were orally administered for 20 days. Distilled water was provided to control group. (**A**) The thymic and splenic indices of normal and CY-treated immunocompromised mice. (**B**) The body weights of normal and CY-treated immunocompromised mice. The results are shown as the mean ± SD (*n* ≥ 8 animals in each group). Significance of the results was determined using one-way analysis of variance followed by Dunnett’s *t*-test (# *p* < 0.05 vs. normal control, * *p* < 0.05, ** *p* < 0.01 and *** *p* < 0.001 vs. CY-control).

**Figure 4 microorganisms-10-02321-f004:**
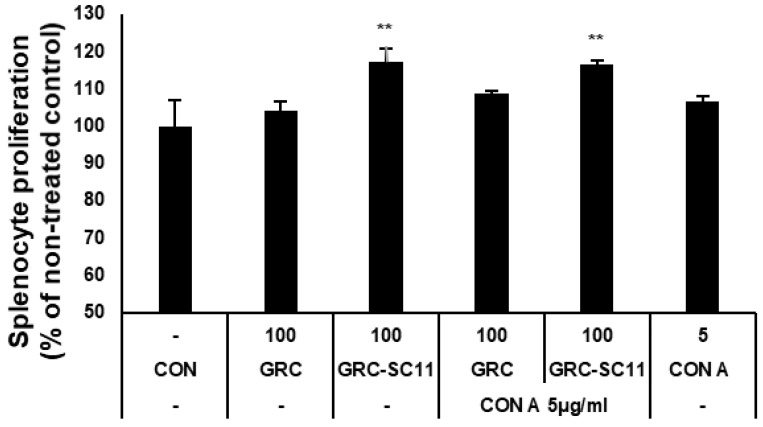
The primary proliferation of splenocyte isolated from CY-treated immunocompromised mice. Con A is concanavalin A as a T-cell mitogen. The results are shown as the mean ± SD (*n* ≥ 8 animals in each group). Data comparisons among groups were analyzed using one-way analysis of variance, followed by Dunnett’s *t*-test (** *p* < 0.01 vs. con).

**Figure 5 microorganisms-10-02321-f005:**
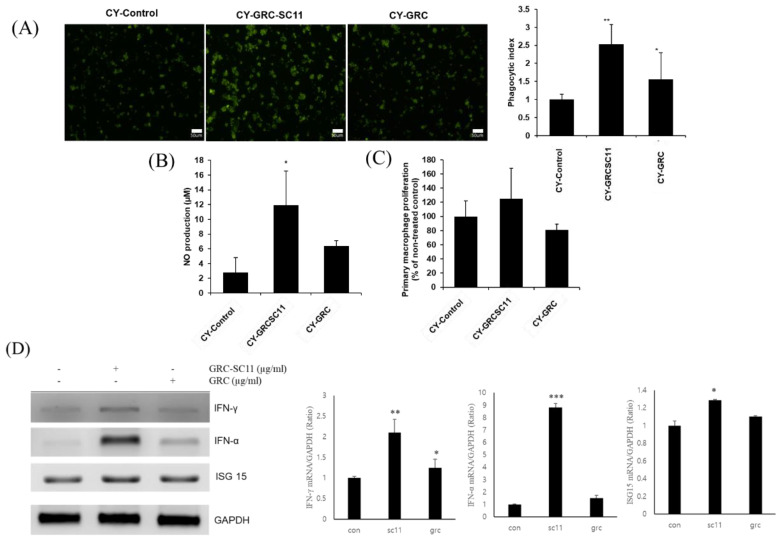
Effects of GRC-SC11 on the phagocytic activity of peritoneal macrophages isolated from CY-treated immunocompromised mice. (**A**) Immunofluorescence images (left panel) of primary peritoneal macrophages from CY-treated immunocompromised mice orally administrated with distilled water, 1% GRC-SC11, and 1% GRC. Quantification of fluorescent beads up-taken by primary peritoneal macrophages from CY-treated immunocompromised mice (exposure for 48 h) (right panel). (**B**) Nitric oxide (NO) production of primary peritoneal macrophages isolated from CY-treated immunocompromised mice. Nitrite levels in the culture media were determined using Griess reagent and were presumed to reflect NO levels. (**C**) Cell proliferation of primary peritoneal macrophages was measured using a CCK-8 assay. (**D**) The levels of INF-γ, IFN-α, and ISG15 mRNA expression in RAW 264.7 cells. One-way analysis of variance was used to compare group means, followed by Dunnett’s *t*-test for the significance of individual comparisons (* *p* < 0.05, ** *p* < 0.01 and *** *p* < 0.001 vs. control group). Each figure is representative of three independent experiments: Scale bars = 50 µm.

**Table 1 microorganisms-10-02321-t001:** Comparison of bioactive compound contents between GRC-SC11 and GRC.

Bioactive Compound	GRC-SC11	GRC
Cordycepin (mg/g)	10.45 ± 2.73 **	6.35 ± 1.64
β-glucan (g/100 g)	28.78 ± 0.31 **	11.83 ± 0.82
Total Flavonoid Content (mg QE/g extract)	6.83 ± 1.98 **	2.8 ± 0.85

** = *p* < 0.05 vs. GRC.

## Data Availability

Not applicable.

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
