# Peer review of "Immunostimulatory Activity of Cordyceps militaris Fermented with Pediococcus pentosaceus SC11 Isolated from a Salted Small Octopus in Cyclophosphamide-Induced Immunocompromised Mice and Its Inhibitory Activity against SARS-CoV 3CL Protease"

_microorganisms, 2022, doi:10.3390/microorganisms10122321_

Round 1
Reviewer 1 Report
In this study the authors have come up with an affordable and easily accessible approach to utilize and enhance the immune stimulating benefits of GRC and have also demonstrated the benefit of this under immunocompromised conditions since that is a major issue during the pandemic where a lot of immunocompromised people have been affected more severely and have to live under more fear. The author show that GRC-Sc11 is capable of preventing replication of SARS-CoV-2 which would be very useful during the continuing pandemic specially if it is an affordable option.
The authors have also done a great job in the discussion to explain their rationale and results to the readers.
Suggestions:
1. Please add the statistical significance * to all their graphs and figures.
2. For the cytokine production, did the authors test other cytokines such as TNF or the downstream genes to IFNY such as the ISGs to have a better understanding of the changes to the cytokines landscape.
Author Response
Dear Reviewers
We greatly appreciate the efforts of the Editorial Board and reviewers for giving us the valuable comments on our manuscript for publication in Microorganisms. Here, we have enclosed a revised version of manuscript in response to the reviewer comments. Included below is a point-by-point description of our responses to the Editor’s and reviewer’s comments. The revision made in the manuscript was written in blue. We cordially hope that you and the reviewers find this revised manuscript acceptable for publication in Microorganisms.
Sincerely,
Hye-Jin Park

Reviewer 2 Report
Reviewer comments
Manuscript title : Immunostimulatory activity of Cordyceps militaris fermented with Pediococcus pentosaceus SC11 isolated from a salted small octopus in cyclophosphamide-induced immunocompromised mice and its inhibitory activity against SARS-CoV 3CL protease
In the current study, the authors investigated the immune-enhancing and anti-viral effects of germinated 13 Rhynchosia nulubilis (GRC) fermented with Pediococcus pentosaceus SC11 (GRC-SC11) isolated from a 14 salted small octopus. They concluded that y indicates 22 that GRC-SC11 might be a potential therapeutic agent for immunocompromised patients who are 23 vulnerable to SARS-CoV infection. The paper is interesting and stimulating but, the reviewer has few comments
- Introduction: Please write a paragraph about how cyclophosphamide (CY)-result in immune-compromised mice.
- In the results section
§ The figure includes symbols a,b, c above the columns but the legend did not include the indications of these symbols. Please add a calcification of these symbols in the figures legend.
§ Figure 1 b is not clear.
§ Figure 5 (D) showing the levels of INF-γ mRNA expression in RAW 264.7 cells. Why there is western blot bands while it was mentioned that INF-γ was measured by PCR only? Also in the same figure, there are (A) Immunofluorescence images, which was not mentioned before in the material and methods.
- In the discussion section
§ Line 399: “The content of cordycepin of GRC-SC11 (9.02 ± 0.54 mg/g) was 1.4-fold 399 higher than that of GRC (6.35 ± 0.33 mg/g)”. already mentioned in the results so no need to be included in the discussion.
Author Response
Dear Editor
We greatly appreciate the efforts of the Editorial Board and reviewers for giving us the valuable comments on our manuscript for publication in Microorganisms. Here, we have enclosed a revised version of manuscript in response to the reviewer comments. Included below is a point-by-point description of our responses to the Editor’s and reviewer’s comments. The revision made in the manuscript was written in blue. We cordially hope that you and the reviewers find this revised manuscript acceptable for publication in Microorganisms.
Sincerely,
Hye-Jin Park

Reviewer 3 Report
The manuscript is a good work and worth publishing in Microorganisms. There are few issues that must be addressed before final publication.
my recommendation is major revision.
1. There is no rationale of using Cordyceps militaris and 2 Pediococcus pentosaceus SC11 against 3CLpro. can the author add some comment on this.
2. how 3clpro was selected? why no the other targets?
3. I recommend the addition of 2D structure of the bioactive compounds.
4. Can the author add molecular docking of the bioactive compounds against 3CLpro.
5. the docking protocol can be also tested against SARS-CoV-2 maine protease (3CLpro).
6. the conlusions should add the future potential of these compounds against the emerging viral and other infections.
7. There are some studies which report the activity of flavoniods. i suggest may add in the introduction or discussion portion.
https://doi.org/10.1002/ptr.6998
doi: 10.1080/07391102.2020.1779128
Author Response

(The authors gave the same response as above.)

Round 2
Reviewer 2 Report
Manuscript has been sufficiently improved to warrant publication in Microorganisms.
Reviewer 3 Report
The paper is properly revised, should be published in its current form